# Comparative study of the unbinding process of some HTLV-1 protease inhibitors using unbiased molecular dynamics simulations

**Fereshteh Noroozi Tiyoula, Hassan Aryapour●\*, Mostafa Javaheri Moghadam**

Department of Biology, Faculty of Science, Golestan University, Gorgan, Iran

\* hassan.aryapour@gmail.com

**Data Availability Statement:** All data and analysis files are uploaded to this address (https://zenodo.org/record/5633143#.YZXlftBBzIU) and are available with DIO:10.5281/zenodo.5633143.

## Abstract

The HTLV-1 protease is one of the major antiviral targets to overwhelm this virus. Several research groups have developed protease inhibitors, but none has been successful. In this regard, developing new HTLV-1 protease inhibitors to fix the defects in previous inhibitors may overcome the lack of curative treatment for this oncovirus. Thus, we decided to study the unbinding pathways of the most potent (compound 10, PDB ID 4YDF, Ki = 15 nM) and one of the weakest (compound 9, PDB ID 4YDG, Ki = 7900 nM) protease inhibitors, which are very structurally similar. We conducted 12 successful short and long simulations (totaling 14.8 μs) to unbind the compounds from two monoprotonated (mp) forms of protease using the Supervised Molecular Dynamics (SuMD) without applying any biasing force. The results revealed that Asp32 or Asp32′ in the two forms of mp state similarly exert powerful effects on maintaining both potent and weak inhibitors in the binding pocket of HTLV-1 protease. In the potent inhibitor's unbinding process, His66′ was a great supporter that was absent in the weak inhibitor's unbinding pathway. In contrast, in the weak inhibitor's unbinding process, Trp98/Trp98′ by pi-pi stacking interactions were unfavorable for the stability of the inhibitor in the binding site. In our opinion, these results will assist in designing more potent and effective inhibitors for the HTLV-1 protease.

## Introduction

Human T-cell leukemia virus type 1 (HTLV-1) was discovered in 1980 as the first oncogenic retrovirus in the project "War on Cancer" in the United States [1]. According to the latest information, 5–10 million people are infected with this virus worldwide, and only 0.25–5% of them are affected by Adult T-cell Leukemia/Lymphoma (ATLL) and HTLV-1-associated myelopathy/tropical spastic paraparesis (HAM/TSP) [2], and also HTLV-1-associated ocular diseases. These diseases are known as HTLV-1 uveitis (HU) and ATL-related ocular [3]. Indeed, the reported numbers are not terrible, but there is no standard treatment for all types of diseases [4]. In addition, only a few regions were evaluated, and many unknown infected people could transmit the virus [5]. So, even low-risk areas are in danger because of Global Village. After HTLV-1 discovery, all its components were identified gradually, and its protease was

**Funding:** Yes Golestan university. The funders had no role in study design, data collection and analysis, decision to publish, or preparation of the manuscript.

**Competing interests:** The authors have declared that no competing interests exist.

discovered in 1989 [6]. HTLV-1 protease is a homodimer protein containing 125 residues in each subunit, which is one of the A2 family of aspartic proteases, with two critical aspartates in the catalytic dyad. This enzyme is essential for viral growth because it cleaves the Gag-Pro-Pol-Env polyprotein, a necessary viral replication component [7]. Since this part is vital for the viral life cycle, it is an interesting target for HTLV-1 demise.

Toward this end, many research groups in different countries succeeded in designing and synthesizing various compounds with inhibitory effects in the micromolar to nanomolar ranges [8, 9]. Finally, some German scientists considered the structural similarities between HTLV-1 and HTLV-3 (HIV) and determined the X-ray structure of Indinavir complexed with HTLV-1 protease, which is the only AIDS protease drug that has an inhibitory effect on HTLV-1 protease in low micromolar concentration. Unfortunately, this drug failed to be used to eradicate HTLV-1 [10]. After being frustrated with AIDS drugs, this team, in 2015, succeeded in synthesizing ten inhibitors that contain the most potent nonpeptidic inhibitor of HTLV-1 protease [11].

All reported HTLV-1 protease compounds only remain as inhibitors, and we do not have any specialized FDA-approved drug for inhibiting this virus. It is evident that experimental research alone is not sufficient. *In silico* methods, like unbiased molecular dynamics (UMD), are needed to provide valuable information for rational drug design, which is the primary goal of all researchers in this field. So, various research groups have used *in silico* methods for different studies, involving: carrying out the docking simulation to investigate the substrate-binding cavity of the IDH1 enzyme [12], useing dynamic simulations and binding free energy studies to design new potent inhibitors against CDK2 [13], prediction of the binding pocket in the interface of the aurora-A-TPX2 complex using molecular docking [14], finding specific naturally originated antagonist of the benzodiazepine binding site by performing docking experiments and molecular mechanics Poisson-Boltzmann surface area analysis [15]. MD simulation offers information about the reaction pathways of the ligand-protein complexes, and it has been considered by many research groups over these years and led to effective drug design [16, 17]. Therefore, besides the importance of one particular drug's binding affinity to a target protein in the traditional drug design, the binding and unbinding processes and the residence time of the compound that interacts with the protein in each intermediate state are just as important. So by a complete understanding of the unbinding mechanism, we can uncover the key elements in the protein-ligand complex interactions [18], ligand flexibility, and solvation effects that are more critical in the rational drug design. The vital information will ultimately appear in a scenario with fully atomistic details [19]. For investigating unbinding pathways of inhibitors, some advanced MD simulation approaches like metadynamics and supervised metadynamics (suMetaD) simulation have been used before [20, 21], and one of the newest MD approaches is the supervised molecular dynamics (SuMD) method. This method performs simulation in replicas with fixed parameters. It gets information regarding metastable intermediate ligand-bound states using a tabu-like supervision algorithm. It is possible to fully unbind small molecules from their molecular targets without any biasing force or potential. In this regard, the SuMD has been utilized to discover the reaction pathways of various ligands in molecular targets [22].

The previous work of our team attempted to achieve Indinavir's interaction with HIV protease and HTLV-1 protease as a comparison test to better understand the complexity of these key proteins [23]. In the following, we decided to examine the unbinding pathways of the most potent and one of the weakest HTLV-1 protease inhibitors retrieved from the last designed compounds using SuMD. In this study, the simulations will reveal the dynamic behavior of HTLV-1 protease as well as the strengths and weaknesses of the selected inhibitor. So, the results will guide the design of new potent HTLV-1 protease inhibitors. Also, this unbiased

comparative study caused monitoring of all important factors in the two different inhibitor-protein interactions. So the outcomes were more reliable.

## Methods

The X-ray crystallography structures of the HTLV-1 protease-ligand complex (PDB IDs: 4YDG, 4YDF [11]) were obtained from the Protein Data Bank. At first, for protein preparation, all protein missing residues and atoms in 4YDG were remodeled and fixed using UCSF Chimera software [24]. Then, for the ligand preparation, according to the practical information, the nitrogen of the pyrrolidine ring was protonated in both compounds [11] and parameterized by ACEPYPE using default settings (the GAFF atom type and BCC partial charges) [25]. HIV protease is an aspartic protease, and there were no studies on HTLV-1 protease, so we used a pattern from HIV research. In some research, protonation is not applied or not mentioned, or in some research, protonation state is applied in chain A. As we wanted a more complete outcome, we used protonation separately in both chains [26, 27]. So after preparing complexes, each catalytic Asp was considered separately as an ionization state based on the monoprotonated (mp) form of the catalytic dyad Asp32-Asp32′ in the active site [28]. Finally, we constructed our systems in GROMACS 2018 [29] using the OPLS all-atom force field [30] and with the TIP3P water model [31]. The considered holoproteins were centered in a triclinic box with a distance of 1 nm from each edge. The next step was to provide a 150 mM neutral physiological salt concentration, sodium, and chloride ions. Then all systems were relaxed in energy minimization using the steepest descent algorithm and reached Fmax of less than 1000 kJ.mol$^{-1}$.nm$^{-1}$. All covalent bonds by Linear Constraint Solver (LINCS) algorithm were constrained to maintain constant bond lengths [32]. The long-range electrostatic interactions were treated using the Particle Mesh Ewald (PME) method [33]. The cut-off radii for Coulomb and Van der Waals (VdW) short-range interactions were set to 0.9 nm for all systems.

At last, the modified Berendsen (V-rescale) thermostat [34] and Parrinello-Rahman barostat [35] were applied for 100 and 300 ps, respectively, for keeping the system in stable environmental conditions (310 K, 1 Bar). To reach complete unbinds, we performed 12 separate replicas (three replicas for each type of mp form) with fixed duration times using the SuMD method with some modifications [36]. During the simulation, the distance between the center of masses of the ligands and selected residues was monitored until complete unbind occurred. This method is based on a tabu-like supervision algorithm without applying any human or non-human biasing force or potential. Herein, we set the center of mass (COM) of ligands as the first spot and the COM of the catalytic aspartic acids (Asp32, Asp32′) as the second spot and ran all simulations with a time window of 500 ps and a time step of 2 fs. After finishing each run, the frame with the longest distance between COMs was selected automatically to extend the next 500 ps simulation. These processes were continued until complete unbind was obtained, which is equal to a distance of 50 Å between the mentioned spots. Finally, all events in every concatenated trajectory file were investigated carefully with GROMACS utilities for data analysis. To picture the important interactions, we used UCSF chimera and used Daniel's XL Toolbox (v7.3.4) to create plots [37], and using Matplotlib to show free energy landscapes (S1 Fig) [38]. The free energy landscapes plots were made base on three variables time, ligand RMSD, and protein RMSD. The ligand and protein RMSD values were selected because they were meaning full and had sharp changes as a function of time during unbindings. Analyzing these plots can reveal the stable states of inhibitors, as well as the residence time of inhibitors in each state over unbinding. Areas that tend to turn blue indicate that the inhibitor has been present in this area for a longer time.

## Results and discussion

Since the only structural difference between compounds 9 and 10 is in the amino and nitro groups on the benzene ring (Fig 1A and 1B), compound 10 ($K_i$ = 15 nM) is approximately 526 times more potent in complex with HTLV-1 protease [11]. Therefore, a proper understanding of the unbinding pathways of these compounds is vital to unveiling secrets that a minor structural difference can have a dramatic effect on inhibitory effects. Therefore, understanding the unbinding process of these inhibitors may provide an insight into the design of next-generation inhibitors.

For a complete understanding of unbinding mechanisms of these two compounds, it is better to get more familiar with this less-known virus′s protease structure and inhibitors features

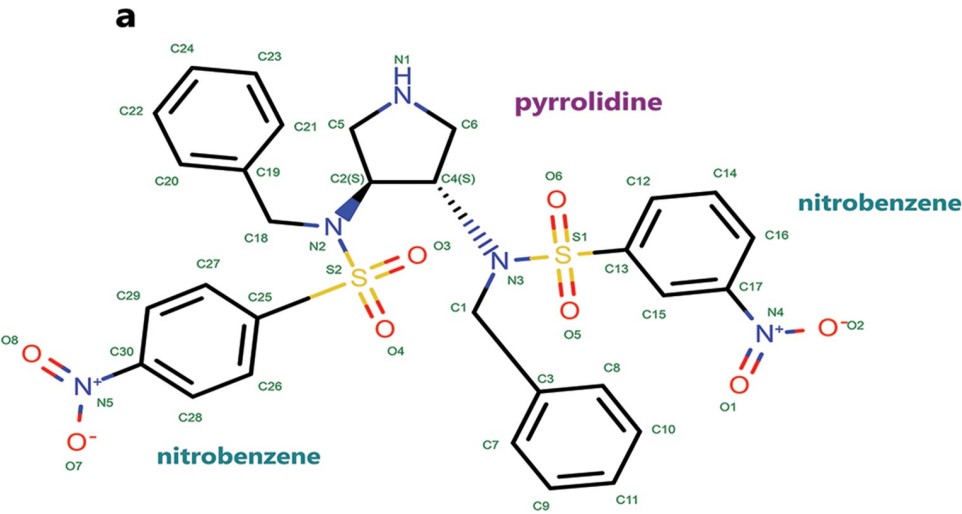

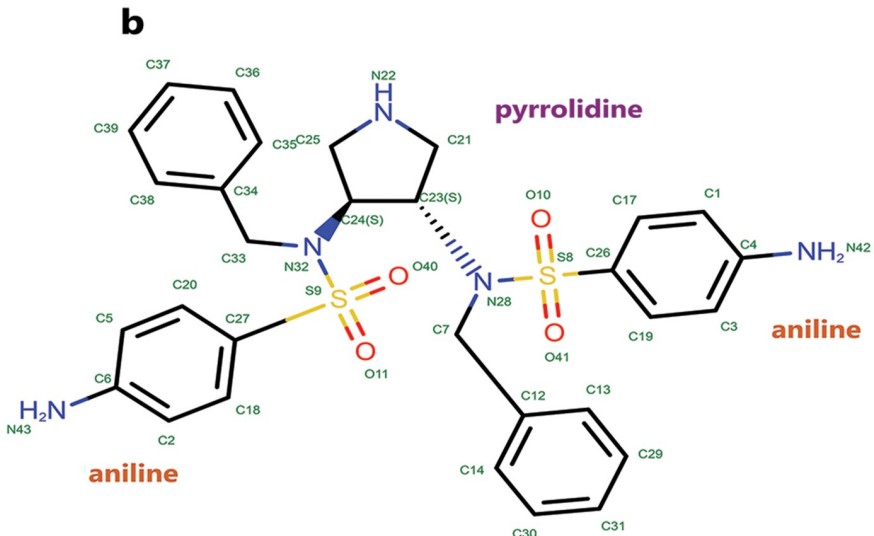

**Fig 1. The 2D structures of selected compounds were obtained from PDB. A**, Compound 9, the inhibitor in (PDB ID 4YDF). **B**, Compound 10, the inhibitor in (PDB ID 4YDG).

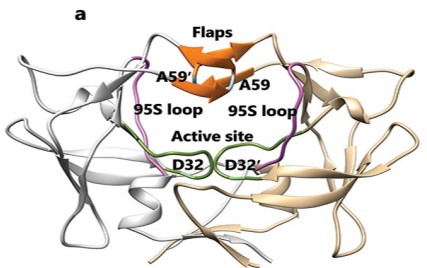
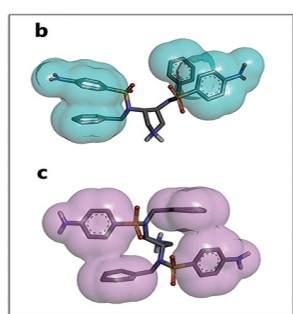

**Fig 2. The 3D structure of HTLV-1 protease (PDB ID 4YDF) and intramolecular interactions of inhibitors. A**, All important domains of HTLV-1 protease: the green area is the active site region, the purple area is the lateral loops or 95S loops part of protease, the orange area is the flaps region, and the blue area is the flap tips part. **B**, Compound 9. **C**, Compound 10.

at first. This homodimer protein has some particular regions with strategic effects in keeping ligands in the protein's binding pocket that are obtained from the analysis of trajectories. The active site region (Leu31-Val39 and Leu31′-Val39′) contains catalytic dyad aspartate residues (Asp32, Asp32′) that are so important in protein-inhibitor interactions. The second essential region is the flaps (Val56-Thr63 and Val56′- Thr63′). The specific residues of Ala59-59′ consider as flap tips in the region of the flaps. Finally, Lateral Loops or 95S loops part of protease (Lys95-Gly102 and Lys95′- Gly102′) are other key regions in this aspartic protease (Fig 2A). For the inhibitors, both compounds have pi-pi intramolecular interactions. With more details, in compound 9, the nitrobenzene ring can form face-to-face pi-pi interaction with the benzene ring (Fig 2B), and in compound 10, the aniline ring can form T-shaped edge-to-face pi-pi interaction with the benzene ring (Fig 2C).

As mentioned, one of the essential parts of this protein is the region of the flaps, which showed high flexibility during simulations (Fig 3E–3H). So, during our simulations, four modes were observed for the flaps. Herein, we considered two factors to show these modes: the distance between COMs of Ala59 and Ala59′ (d1) and the second one is the distance between COMs of Ala59′ and Asp32′ (d2). The second factor can be even between COMs of Ala59 and Asp 32 due to flaps′ handedness opening. In the close form, the maximum amount of d1 and d2 are 10 and 15 Å, respectively (Fig 3A). In the semi-open form, the maximum amount of d1 and d2 is 14 and 20 Å, respectively (Fig 3B). In the open state, the minimum amount of d1 is 14 Å, and the maximum amounts of d1 and d2 are 20 Å (Fig 3C). In the wide-open form, d1 and d2 must be more than 20 Å (Fig 3D).

For our purpose, we had various simulations, that in total, we could have 12 successful unbindings (a total of 14.800 μs) for the two compounds from a minimum of 94 ns to a maximum of 4.4 μs in both mp forms. At last, for providing comprehensive information, all the events in each frame of trajectories were investigated carefully, and different analyses were performed on them.

Like the weak compound, we had two mp forms of the potent compound (AspH32 and AspH32′). In this regard, in the duration times of 4.4 μs (Fig 4A) and 260 and 305 ns (Fig 4B), which were in the chain A, Asp32 protonated state, we saw a uniform mechanism to unbind with some essential differences that caused a significant difference in one of simulation time. So, in the first state of rep1, 2, and 3 (Fig 4C–4E), Asp32′, which had salt bridge interaction with the positive charge of the pyrrolidine ring, plays a crucial role in preserving ligand in the binding pocket of protease. This acidic residue is essential because it is located almost at the bottom and center of the binding pocket. This residue considers a strategic residue due to the

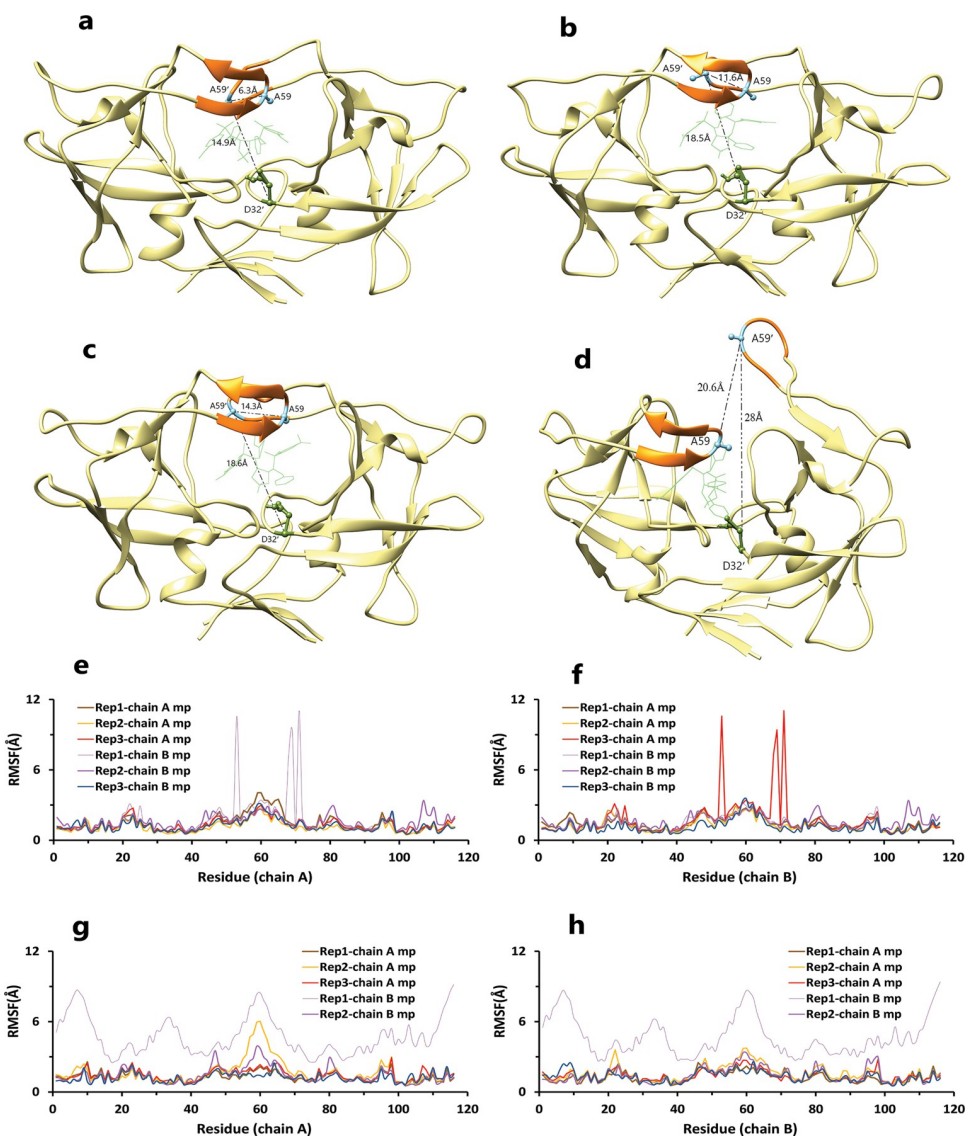

**Fig 3. Different modes of the flap. A**, Close form of the flap. **B**, Semi-open form of the flap. **C**, Open form of the flap. **D**, Wide-open form of the flap. **E, F**, RMSF values of HTLV-1 protease in the 4YDG PDB code, during simulations. **G, H**, RMSF values of HTLV-1 protease in the 4YDF PDB code during simulations.

positive charge of pyrrolidine. Parallel to that, His66′ by cation–pi interaction [39, 40] with an aniline ring was the second important preserving factor. In addition, Ala59′ in the flap tip by forming H-bond with the atom of $O_{10}$ (Fig 4H) and also Ala35′ by VdW interactions with a benzene ring, Asp36′ by forming H-bond with an aniline fragment in the active site, and finally Ile100′ in 95S loop (Fig 4J–4L), with VdW interaction, blocked all the exit routes, like the fence (Fig 4F). As mentioned before, His66′ was the second essential residue in this state, which was a supporter of Asp32′ to fix the inhibitor in the binding pocket. As time passing, Asp32′ loosed its superpower of preserving, and the inhibitor entered the second intermediate state. In this state, Lys95′, forming a hydrogen bond with the atom of $O_{10}$, along with His66′ cation-pi interaction with the benzene ring, was a third essential residue. This residue increased protein-ligand interactions time in the rep1 and was absent in rep2 and rep3 (Fig 4G). According to

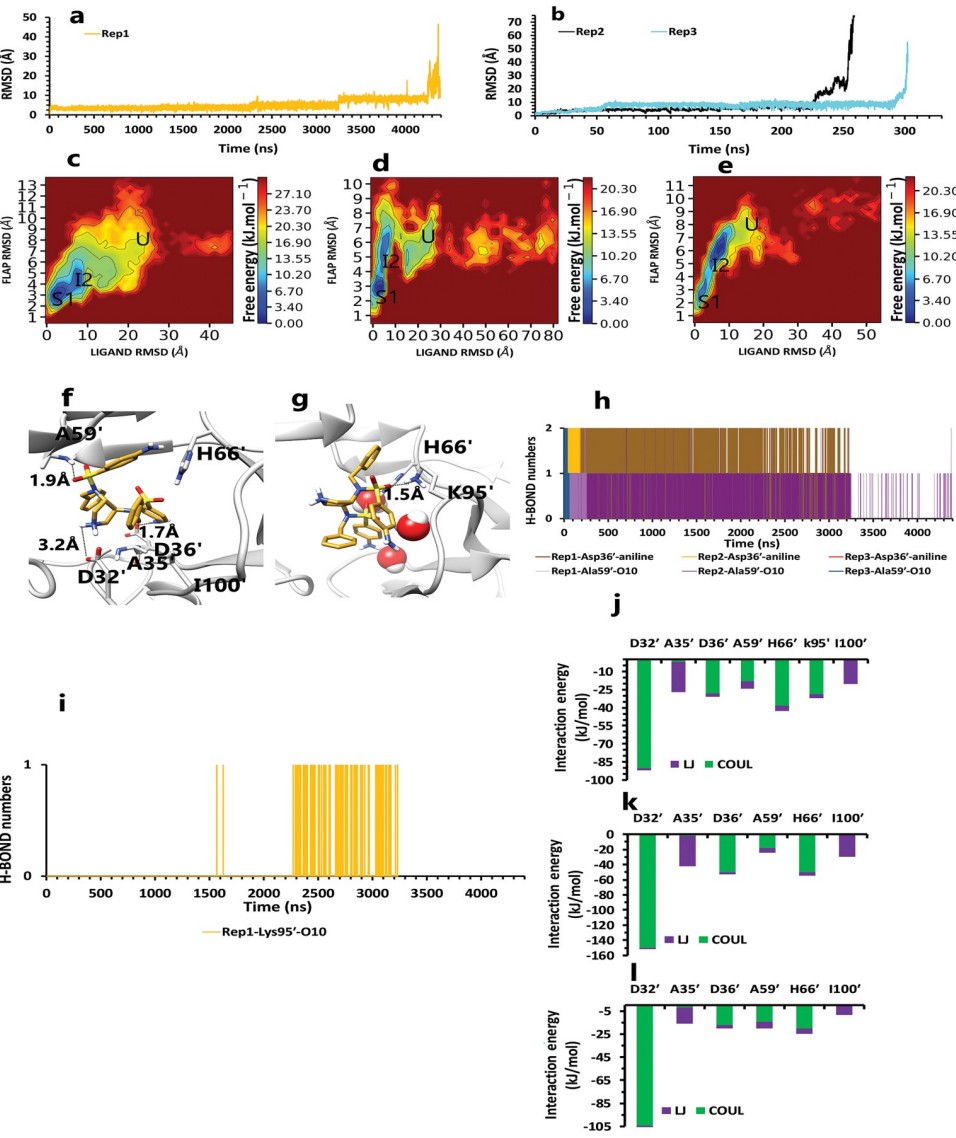

**Fig 4. The details of compound 10 unbinding pathways in complex with HTLV-1 protease when Asp32 of chain A was protonated in three replicas. A**, RMSD value of the ligand from binding pose to complete unbinding in the rep1. **B**, RMSD values of the ligand from binding pose to complete unbinding in the rep2 and rep3. **C**, **D**, and **E**, The free energy landscape of rep1, 2, and 3 during the unbinding process (state (S), intermediate state (I), unbound (U)), respectively, which was calculated by using "gmx sham". **F**, The interactions between the ligand and essential residues in the binding pose of rep1, 2, and 3. **G**, The new interactions between the inhibitor and particular residues in the second intermediate state of rep1. **H**, Hydrogen bond numbers of Asp36′ and Ala59′ with the inhibitor in rep1, 2, and 3. **I**, Hydrogen bond numbers of Lys95′ with the inhibitor in rep1. **J**, **K**, and **L**, The average of most important interaction energies of the protein-ligand complex in rep1, 2, and 3, respectively.

significant differences in replicas simulation times, the effects of the Lys95′ hydrogen bond (Fig 4I) appear more pronounced. Finally, ligand pi-pi intramolecular interactions, observed during the whole time of simulations (Fig 5A–5C), slowly weakened all-important protein-inhibitor interactions. The critical point was that, over the entire simulation time, flaps positioning impacted the ligand's behaviors, so the exit process started when the flaps began to open, and Ala59′ loosed its effect (Fig 5D–5F) in the second intermediate state gradually with the help of water mediation (Fig 6A–6C).

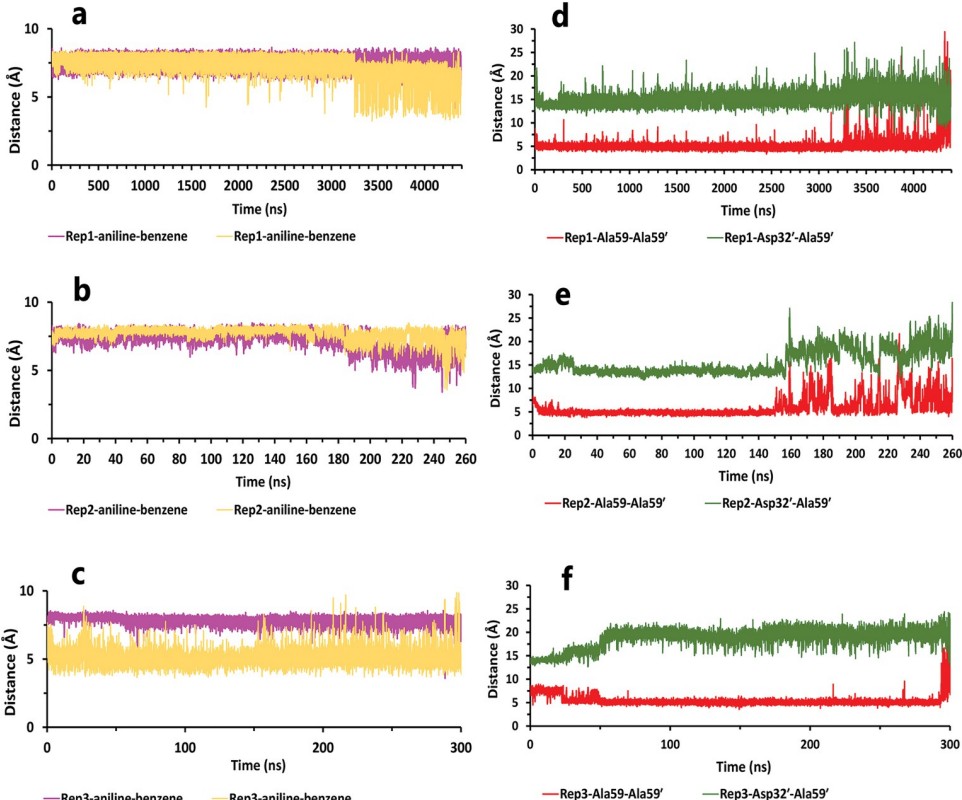

**Fig 5. The details of distances between particular parts of compound 10 complexed with HTLV-1 protease when Asp32 of chain A was protonated in three replicas. A**, **B**, and **C**, The distance between COMs of both aniline rings and benzene rings, which were in a position that could form pi-pi intramolecular interactions in all replicas. These plots prove that these fragments were so close together during the simulation. **D**, **E**, and **F**, The distance between COMs of Ala59 and Ala59′, and also Asp32′ and Ala59′ in all replicas (these plots should be checked along with Fig 3).

Conversely, in the other state of protonation (AspH32′), we saw a uniform pathway that was dissimilar to the previous unbinding proccess with the different lengths of times involving: 94, 320, and 790 ns (Fig 7A and 7B). In the first state (S1 Fig) of these pathways (Fig 7C–7E), Asp32 was as important as expected. Asp36 and Asp36′, Leu57, and Ala59′ are the residues that acted as auxiliary agents (Fig 7J–7L) to the pivotal amino acid (Asp32). At the first simulation times, along with the salt bridge of Asp32 and pyrrolidine fragment (Fig 7F), both aniline rings had H-bonds with Asp36 and Asp36′ in the active site (Fig 7H). Along with these residues, Leu57 and Ala59′ formed a hydrogen bond with an aniline fragment and $O_{41}$ atom of inhibitor, respectively (Fig 7G and 7I). In the following, in the lack of His66 and Lys95 effects, after time passing with the help of pi-pi ligand intramolecular interactions and water molecules effect (Fig 8A–8C), active site and flaps' important residues lost their effects, and full unbind was observed between the flaps (Figs 8D–8F and 9A–9C).

Depending on close and open flaps and mp states, we had different unbinding mechanisms for the weaker inhibitor. Accordingly, compound 9 was unbound in 148 ns (Fig 10A), 3.5 μs, and 3 μs (Fig 10B) when Asp 32 of chain A was protonated. In the first state (S1 Fig) (Fig 10C) of the rapid unbinding pathway (rep1), a repulsive force occurred between the pyrrolidine ring of ligand and AspH32 of the binding pocket, and because of no attractive interactions in this area, AspH32 forced the ligand to push out. In addition to this interaction, VdW interaction between both nitrobenzene rings and one of the benzene rings of inhibitor and Leu57,

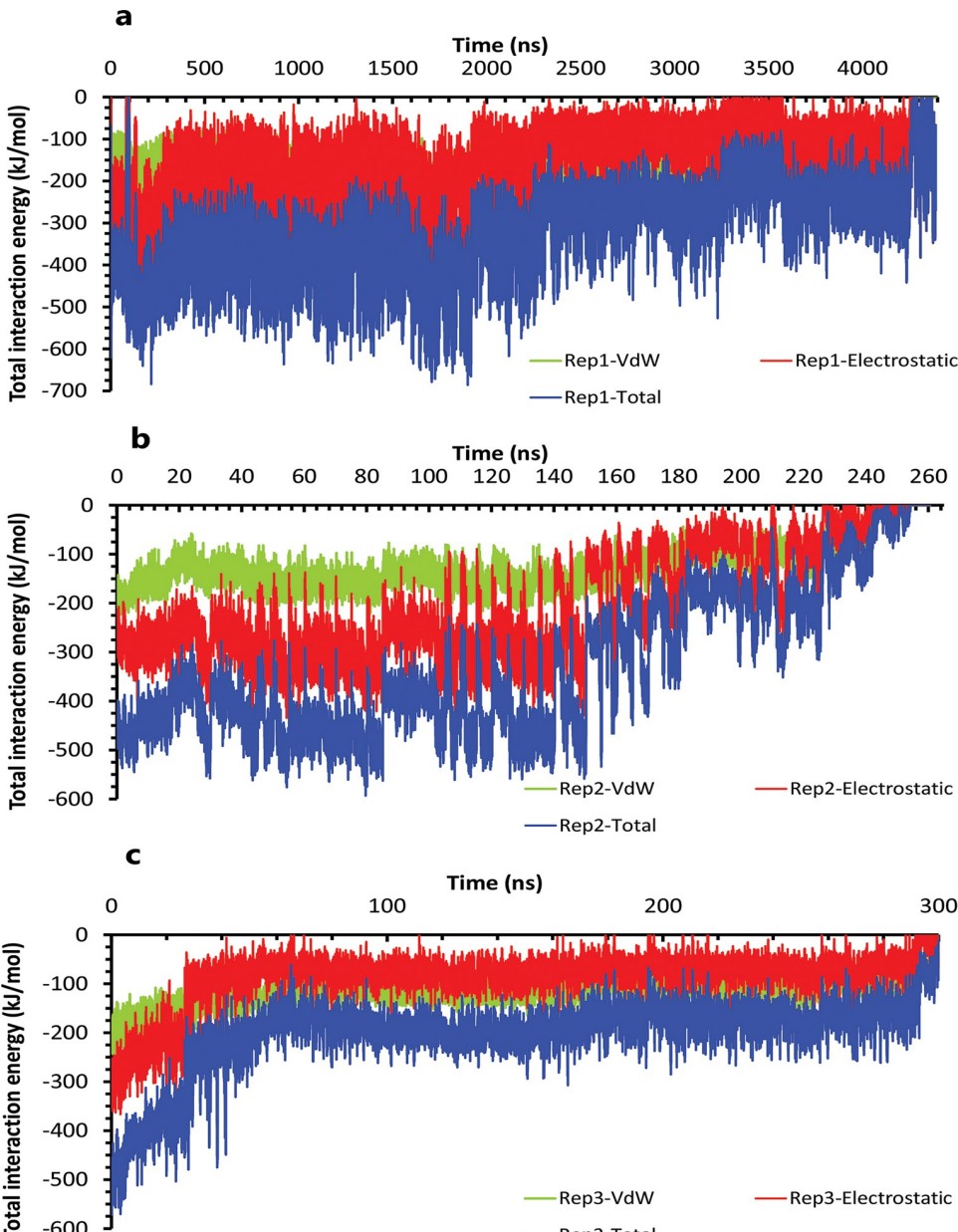

**Fig 6. The interaction energies plots of compound 10 in complex with HTLV-1 protease when Asp32 of chain A was protonated in three replicas. A**, **B**, and **C**, The total VdW and electrostatic interactions energies of protein-inhibitor complexes in rep1, 2, and 3.

Gly58, and Ala59 in the close flap region and pi-pi stacking interaction of Trp98′ and nitrobenzene fragment and also pi-alkyl interaction of Ile100′ with the benzene ring, were other protein-inhibitor significant interactions, which were not potent enough to prevent from repulsive interaction effect (Fig 10F and 10K). In the two other longer simulations, compound 9 was unbound in 3.5 μs in wide-open flaps (rep2) and 3 μs in close and semi-open flaps (rep3).

Similarly, Asp32′ was the most important amino acid with its salt bridge and the only common point in both pathways. In the first state of rep2 (S1 Fig) (Fig 10D), due to handedness

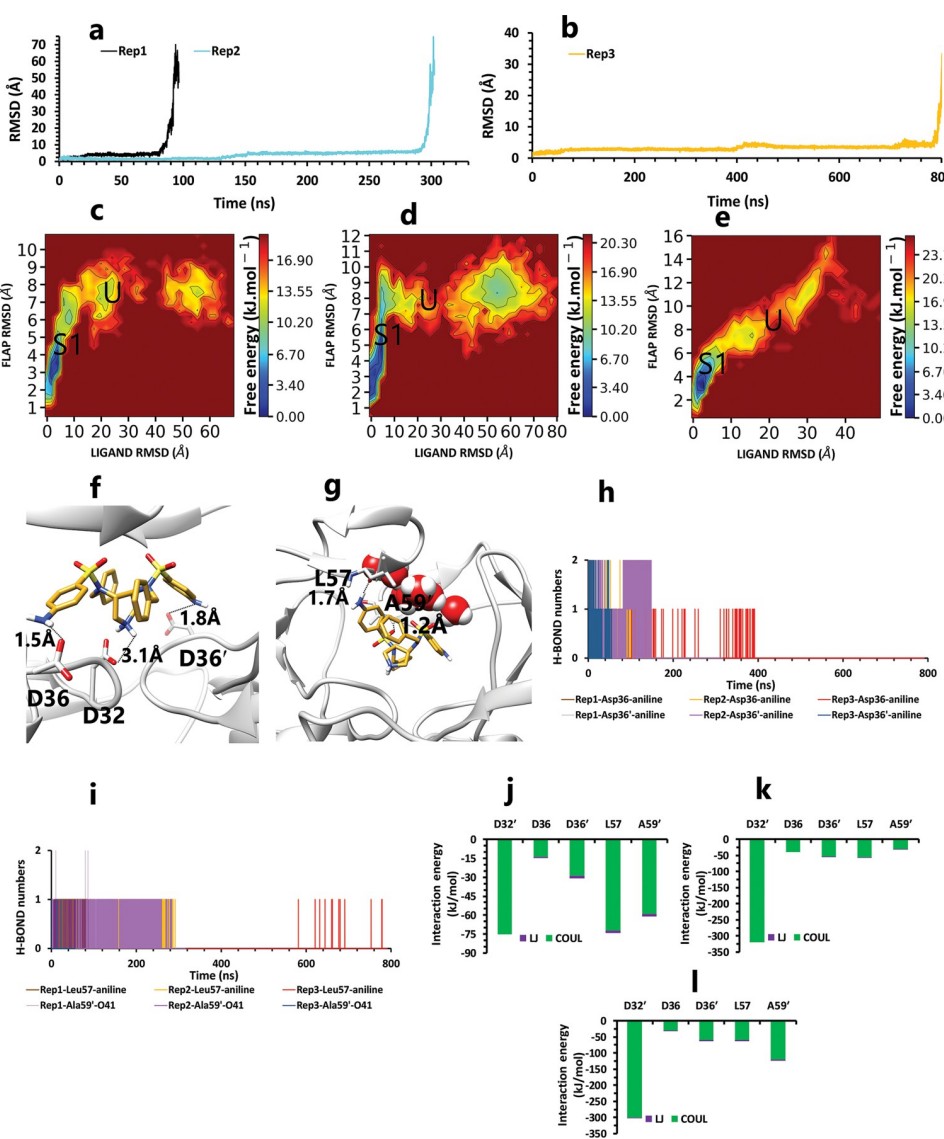

**Fig 7. The details of compound 10 unbinding pathways in complex with HTLV-1 protease when Asp32′ of chain B was protonated in three replicas. A**, RMSD values of the ligand from binding pose to complete unbinding in the rep1 and rep2. **B**, RMSD value of the ligand from binding pose to complete unbinding in the rep3. **C**, **D**, and **E**, The free energy landscape of rep1, 2, and 3 during the unbinding process (state (S), intermediate state (I), unbound (U)), respectively, which was calculated by using "gmx sham". **F**, The interactions between the ligand and essential residues of the active site in the binding pose of rep1, 2, and 3 **G**, The interactions between the ligand and essential residues of the region of the flaps in the binding pose of rep1, 2, and 3. **H**, Hydrogen bond numbers of Asp36 and Asp36′ with aniline fragment in rep1, 2, and 3. **I**, Hydrogen bond numbers of Leu57 and Ala59′ with the inhibitor in rep1, 2, and 3. **J**, **K** and **L**, The average of most important interaction energies of the protein-ligand complex in rep1, 2, and 3, respectively.

opening, only one of the flaps had forward and backward motions, so Leu57′, Gly58′, Ala59′ by VdW interactions kept the ligand in exposing to Asp32′. Also, in this state, Trp98 in the lateral loop built up pi-pi stacking interaction with the nitrobenzene ring of the ligand, and Trp98′ built up pi-pi stacking interaction with the benzene ring of another side of the inhibitor (Fig 10G and 10L). So even with enough space for the exit, the inhibitor was still in blockage. These important protein-inhibitor interactions were maintained until the effect of the Asp32′

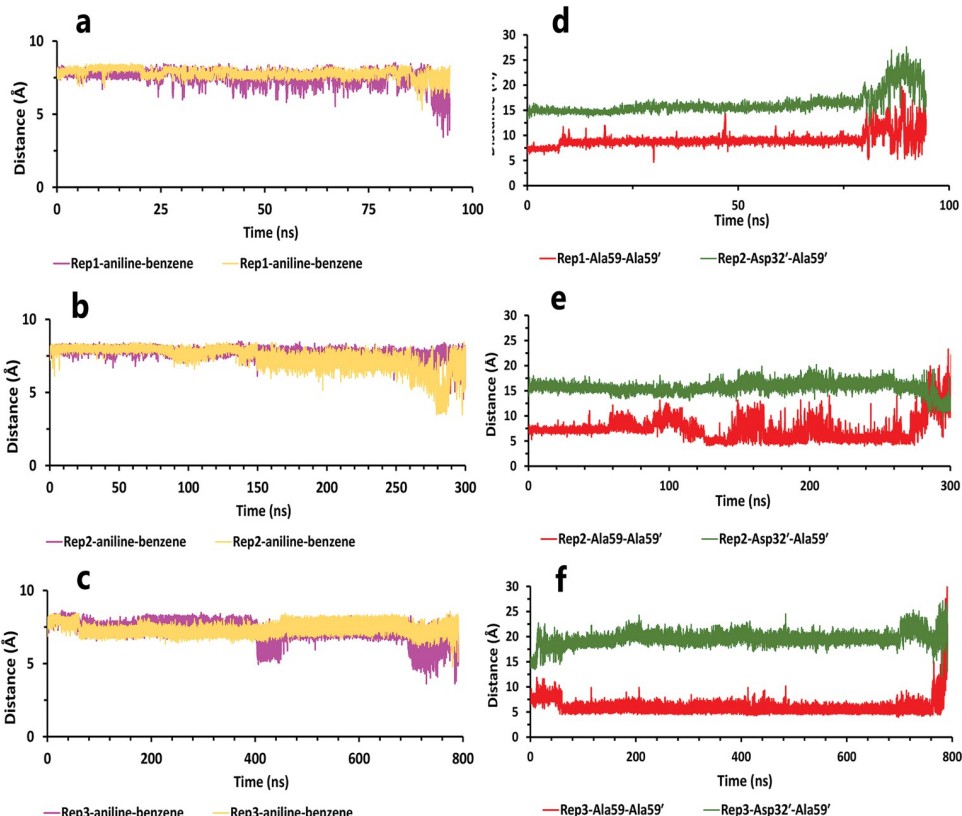

**Fig 8. The details of distances between particular parts in compound 10 in complex with HTLV-1 protease, when Asp32 of chain B was protonated, in three replicas. A**, **B**, and **C**, The distance between COMs of both aniline rings and benzene rings, which were in a position that could form pi-pi intramolecular interactions in all replicas. These plots prove that these fragments were so close together during the simulation. **D**, **E**, and **F**, The distance between COMs of Ala59 and Ala59′ and also Asp32′ Ala59′ in all replicas (these plots should be checked with Fig 3).

became faded, and other agents lost their effect. Unexpectedly, the interesting point was that the complete unbinding process does not occur from the region of the flaps. In the rep3 pathway, that the flaps were close or semi-open the whole time, from the first state (Fig 10E), not only Asp32′ was necessary, and Asp36 in a close position to Asp32′ was powerful too (Fig 10M).

On the other hand, during the first two states, Asp36, by forming pi-anion interaction [41] with nitrobenzene fragment, was momentous as a second ligand preserving residue (Fig 10H), which was promoted to the first important factor in the following intermediate state by replacing pi-anion interaction with the salt bridge with pyrrolidine fragment (Fig 10I). From a holistic view, even though Asp32′ was more critical for protein, it was effective until the second intermediate state or until 2 μs, but Asp36 (Fig 10J) was effective until complete unbind. Actually, the ligand in all replicas showed face-to-face pi-pi intramolecular interactions between mentioned fragments that caused weakened important protein-ligand interactions gradually with the help of the water mediation effect (Fig 11A–11C). Finally, for the flaps behaviors in all replicas, we saw a new opening form for the rep2 as it was opened from chain A (Fig 11E), and for rep1 and rep3 wide opening (Fig 11D and 11F) was not seen until complete unbound (Fig 12A–12C).

On the contrary, when the Asp32 of chain B was protonated, we saw the same mechanism during the 1.2 μs, 410, and 450 ns of simulations (Fig 13A and 13B). In the first state of these

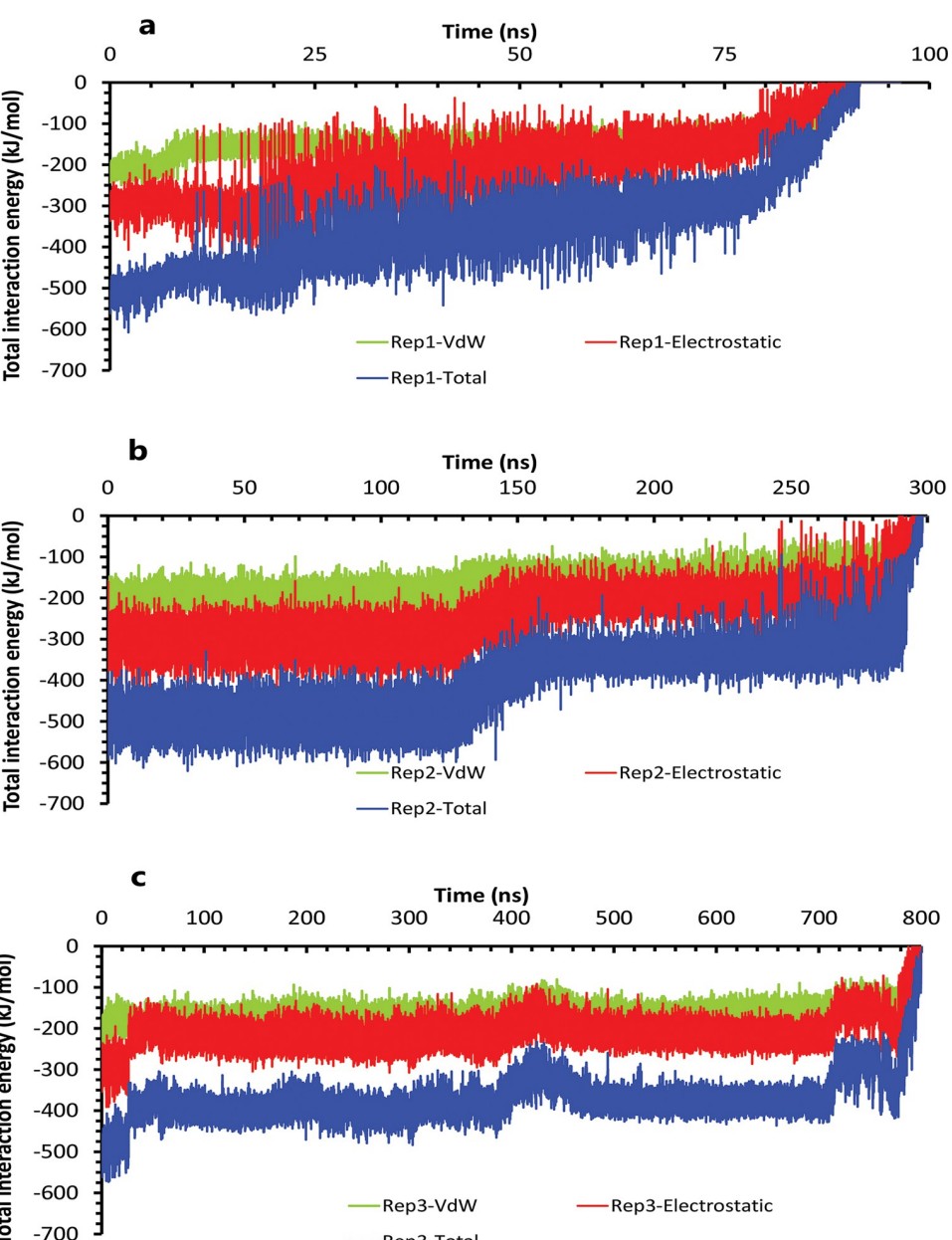

**Fig 9. The interaction energies plots of compound 10 in complex with HTLV-1 protease when Asp32 of chain B was protonated in three replicas. A**, **B**, and **C**, The total VdW and electrostatic interactions energies of protein-inhibitor complexes in rep1, 2, and 3.

replicas (Fig 13C–13E), the ligand was surrounded by interactions of some residues in both chains (Fig 13G–13I). Asp32 had salt bridge interaction with pyrrolidine fragment as a most important interaction. In more detail, this fragment also had VdW interaction with Gly34, one of the nitrobenzene rings was in VdW interactions with Leu57′ and Gly58′ in the flaps regions for the other fragments. The benzene rings were in important interactions involving: pi-pi interaction with Trp98 and pi-alkyl interaction with Ile100 on one side, and pi-pi interactions with Trp98′ and pi-alkyl interaction with Ile100′ on the other side (Fig 13F). It may be due to

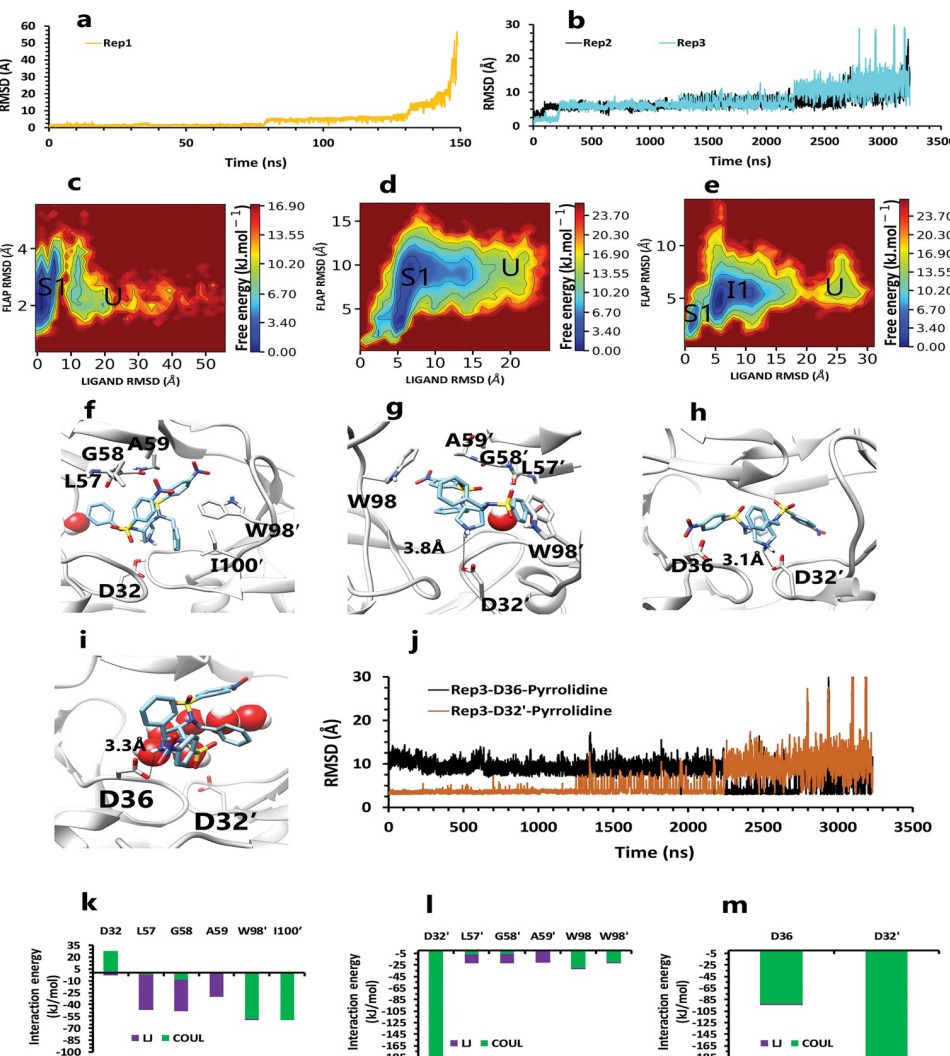

**Fig 10. The details of compound 9 unbinding pathways in complex with HTLV-1 protease when Asp32 of chain A was protonated in three replicas. A**, RMSD value of the ligand from binding pose to complete unbinding in the rep1. **B**, RMSD values of the ligand from binding pose to complete unbinding in the rep2 and rep3. **C, D**, and **E**, The free energy landscape of rep1, 2, and 3 during the unbinding process (state (S), intermediate state (I), unbound (U)), respectively, which was calculated by using "gmx sham". **F**, The interactions between the ligand and essential active site residues in the rep1. **G**, The interactions between the ligand and essential active site residues in the rep2. **H**, The interactions between the ligand and essential active site residues in the binding pose of rep3. **I**, The new interactions between the inhibitor and particular residues in the second intermediate state of rep3. **J**, The distance between COMs of pyrrolidine ring and Asp36 and Asp32′ in rep3, to show after 2us of simulation this fragment get closer to Asp36 and get farther from Asp32′. **K, L**, and **M**, The average of most important interaction energies of the protein-ligand complex in rep1, 2, and 3, respectively.

the high number of important factors; it seems that compound 9 is potent, but except for Asp32 other agents did not have any significant effect. So, they could not keep the ligand after disappearing the Asp32 effect. Thus, as time passed, intramolecular interactions of ligand and water mediation contributed to full unbinding in these three replicas (Fig 14A–14C). Ultimately for the flaps effects, in rep1, the flaps showed high motions, and even though the flaps were wide open (Fig 14D), the full unbind did not occur from this region. In rep2 and 3, the inhibitor unbounded between semi-open flaps forms (Figs 14E, 14F and 15A–15C).

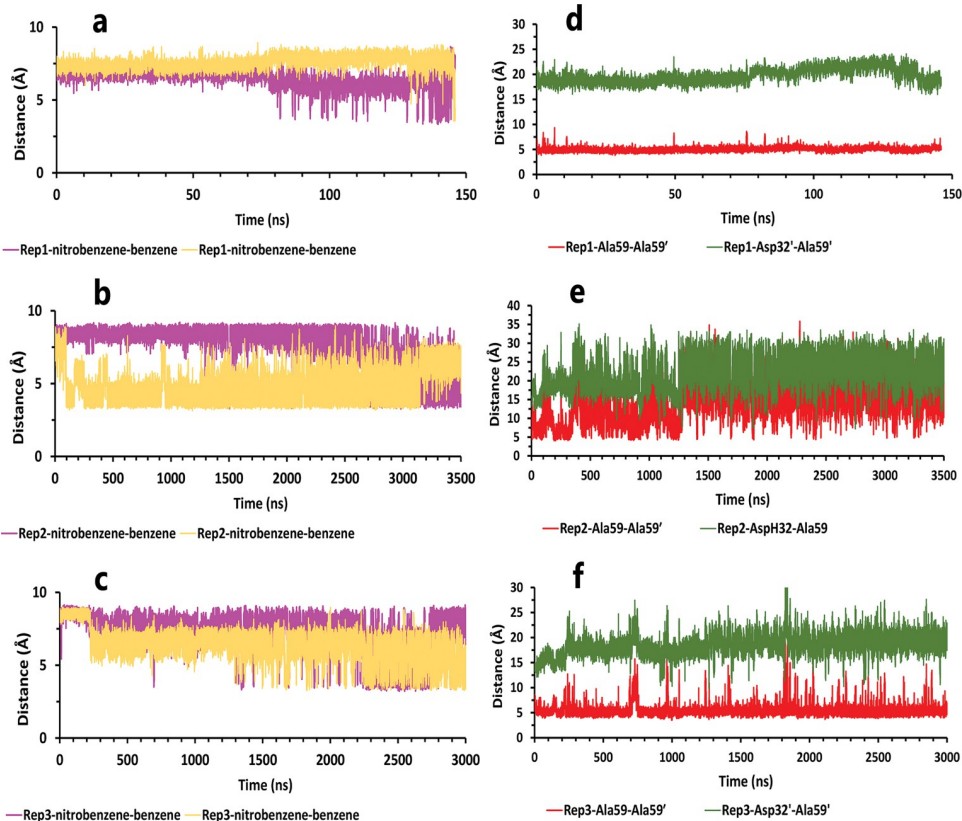

**Fig 11. The details of distances between particular parts in compound 9 in complex with HTLV-1 protease, when Asp32 of chain B was protonated, in three replicas. A**, **B**, and **C**, The distance between COMs of both nitrobenzene rings and benzene rings, which were in a position that could form pi-pi intramolecular interactions in all replicas. These plots prove that these fragments were so close together during the simulation. **D**, The distance between COMs of Ala59 andAla59′ and also Asp32′ Ala59′ in the rep1 (these plots should be checked with Fig 3). **E**, The distance between COMs of Ala59 andAla59′ and also AspH32 Ala59 in the rep2. **F**, The distance between COMs of Ala59 andAla59′ and also Asp32′ Ala59′ in the rep3. **G** and **H**, The total interactions energies of protein-inhibitor complexes in rep1, 2, and 3.

## Conclusion

The atomistic details of unbinding pathways of selected inhibitors in all replicas with various times, the importance of Asp32′ in chain A protonation state and Asp32 in chain B protonation state are pretty straightforward. Due to its strategic position, this effective residue could play a critical role in keeping the ligand in the binding pocket for a long time, so the more exposed to Asp32 or Asp32′, the more inhibitory effects. The pyrrolidine fragment was held well by Asp32 or Asp32′ from the native binding pose of the two compounds, which correlates with experimental research [11]. Thus the interactions of other fragments with other residues in different protein regions caused significant differences.

Herein, we cannot conclude which state of protonation occurs, so with our obtained information for the potent compound in chain A protonation state, His66′ with its cation–pi interaction with an aniline ring of inhibitor was a perfect supporter to Asp32′. This residue's effect was absent in the other form of protonation state and caused a significant difference in simulation time. In the weak inhibitor unbinding pathways, Trp98 and Trp98′ with pi-pi interactions, due to their close position to one of the exit areas, were not good supporters for Asp32 or Asp32′, like His66′. His66′, due to its far position from the bottom and center of the binding

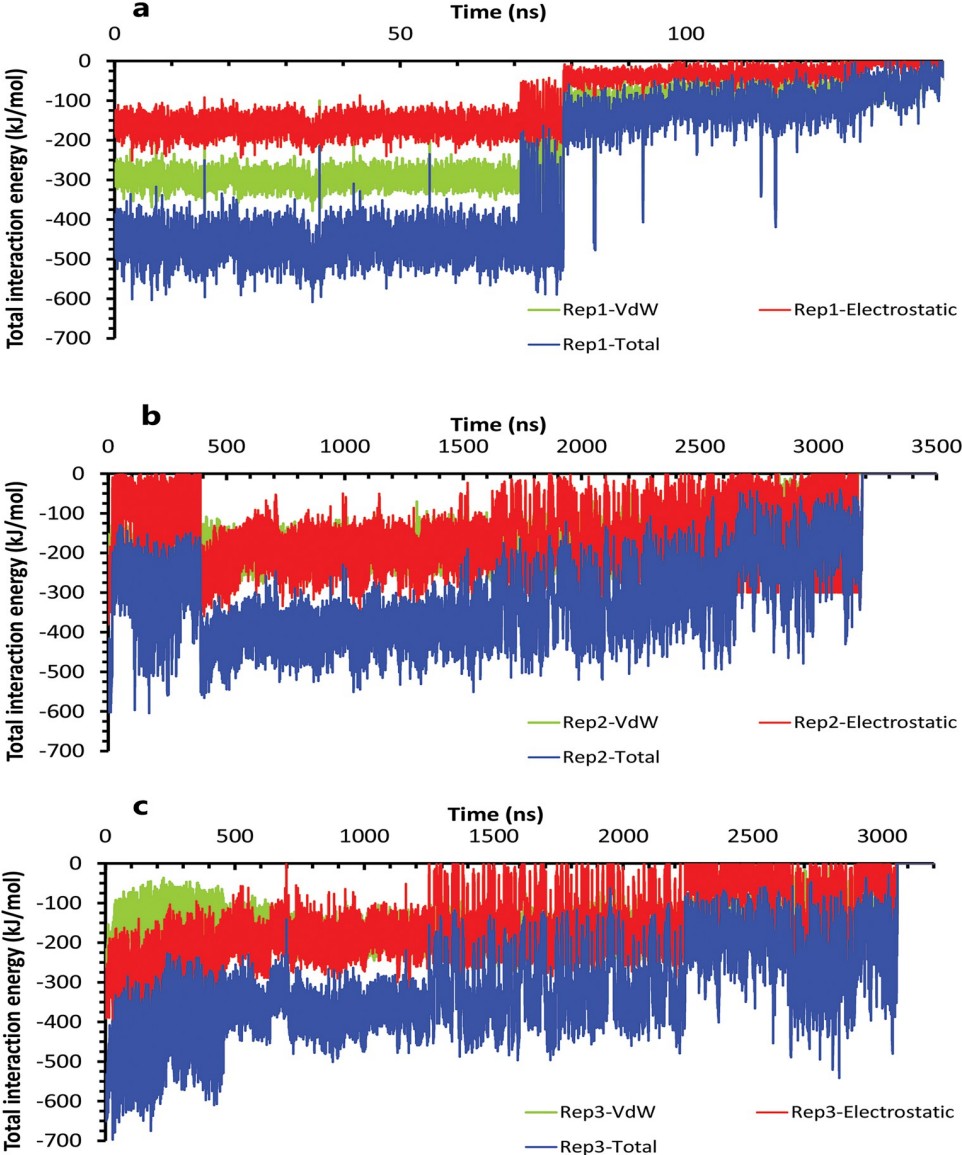

**Fig 12. The interaction energies plots of compound 9 in complex with HTLV-1 protease when Asp32 of chain A was protonated in three replicas. A**, **B**, and **C**, The total VdW and electrostatic interactions energies of protein-inhibitor complexes in rep1, 2, and 3.

pocket, could fix aniline fragments and decrease ligand fluctuations. The two mentioned tryptophan were closer to the essential aspartic acids, and there was enough space for ligand fluctuations. For this reason, Asp36 in the active site that was close to the exit area could be a competitor with Asp32′ and was not a good interaction for keeping the ligand in the binding pocket. Similarly, attenuating effect of Trp98/Trp98′ residues in unbinding pathways of the weak inhibitor correlates with our other research result [23]. These residues' interactions are unfavorable to Indinivar's stability in complex with HTLV-1 protease and result in it being a weak inhibitor. As we said before, both compounds had intramolecular interactions that caused weakening critical protein-ligand interactions as time passing. These two compounds did not have the same intramolecular interactions type, so in the weak inhibitor, face-to-face

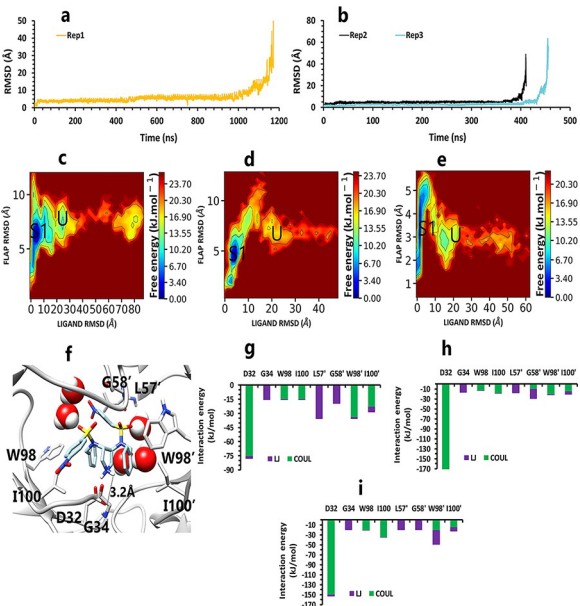

**Fig 13. The details of compound 9 unbinding pathways in complex with HTLV-1 protease when Asp32 of chain B was protonated in three replicas. A**, RMSD value of the ligand from binding pose to complete unbinding in the rep1. **B**, RMSD values of the ligand from binding pose to complete unbinding in the rep2 and rep3. **C**, **D**, and **E**, The free energy landscape of rep1, 2, and 3 during the unbinding process (state (S), intermediate state (I), unbound (U)), respectively, was calculated using "gmx sham". **F**, The interactions between the ligand and essential active site residues in the rep1, 2, and 3. **G**, **H**, and **I**, The average of most important interaction energies of the protein-ligand complex in rep1, 2, and 3, respectively.

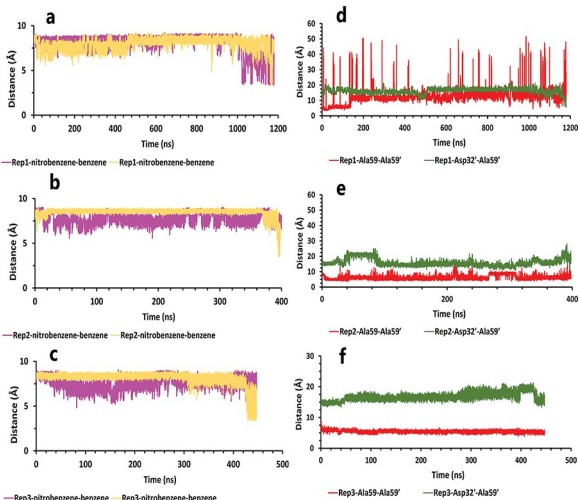

**Fig 14. The details of distances between particular parts of compound 9 complexed with HTLV-1 protease when Asp32 of chain B was protonated in three replicas. A**, **B**, and **C**, The distance between COMs of both nitrobenzene rings and benzene rings, which were in a position that could form pi-pi intramolecular interactions in all replicas. These plots prove that these fragments were so close together during the simulation. **D**, **E**, and **F**, The distance between COMs of Ala59 andAla59′ and also Asp32′ Ala59′ in all replicas (these plots should be checked with Fig 3).

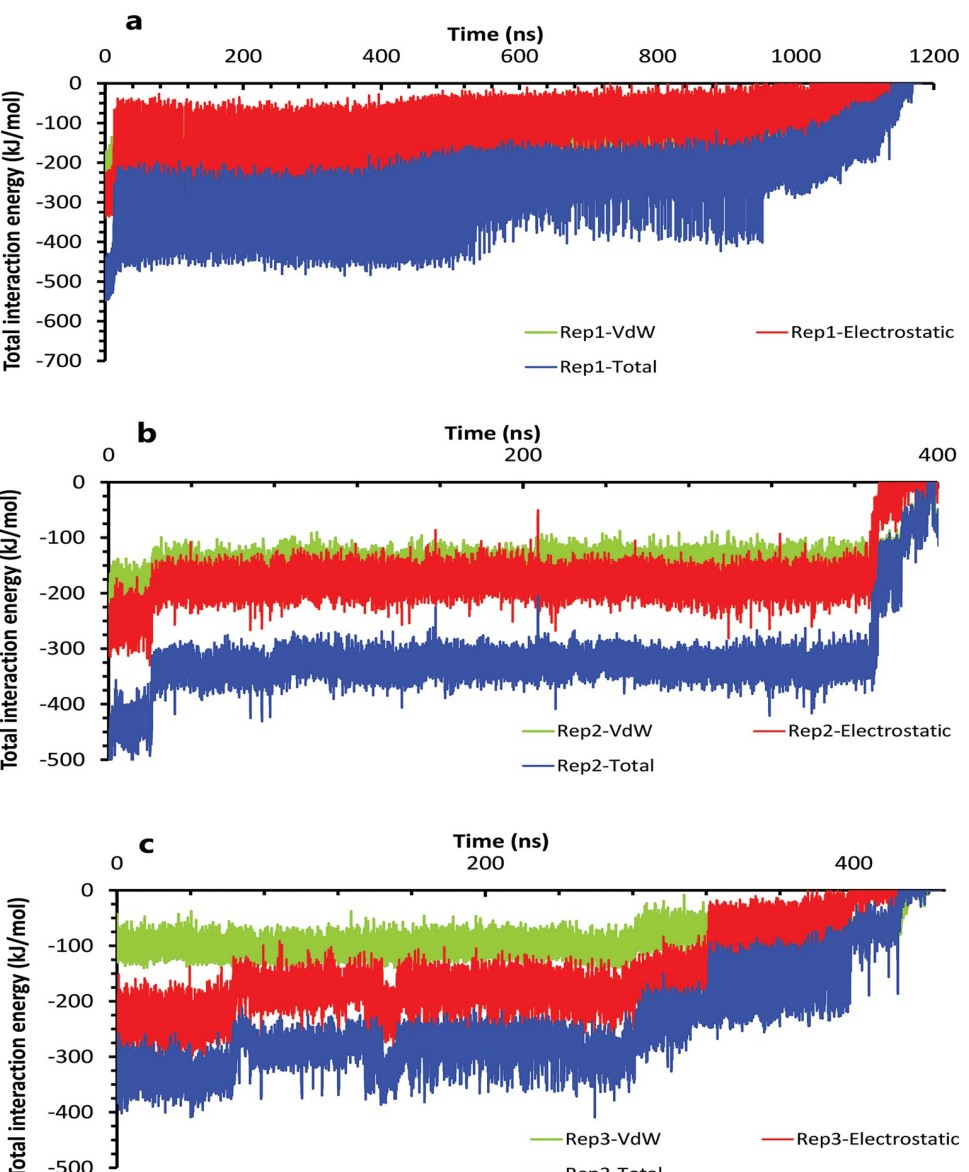

**Fig 15. The interaction energies plots of compound 9 in complex with HTLV-1 protease when Asp32 of chain B was protonated in three replicas. A**, **B**, and **C**, The total VdW and electrostatic interactions energies of protein-inhibitor complexes in rep1, 2, and 3.

pi-pi interactions resulted in the loss of significant pi interactions with the protein. However, the potent inhibitor could have formed more important pi interactions with the protein along with intramolecular interactions. Overall, this obtained information is valuable for designing a new generation of inhibitors against this molecular target.

Because no similar simulation has been performed on the interaction of these inhibitors with HTLV-1 protease, we were only able to compare the results of native states with the crystallographic binding poses. So, except Asp32, Asp32′, in the potent inhibitor and Trp98, and in the weaker inhibitor, Asp36, Trp98 and Trp98′ were the only common critical residues [11].

## Supporting information

**S1 Fig. The free energy landscape plots of all replicas.**
(TIF)

**S2 Fig.**
(TIF)

## Author Contributions

**Conceptualization:** Hassan Aryapour.

**Data curation:** Fereshteh Noroozi Tiyoula, Hassan Aryapour.

**Formal analysis:** Fereshteh Noroozi Tiyoula, Mostafa Javaheri Moghadam.

**Methodology:** Hassan Aryapour, Mostafa Javaheri Moghadam.

**Project administration:** Hassan Aryapour.

**Software:** Mostafa Javaheri Moghadam.

**Supervision:** Hassan Aryapour.

**Validation:** Hassan Aryapour.

**Writing – original draft:** Fereshteh Noroozi Tiyoula.

**Writing – review & editing:** Hassan Aryapour.

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
