## [Decision Letter · Decision Letter 0]

15 Mar 2022

PONE-D-22-00753Comparative study of the unbinding process of some HTLV-1 protease inhibitors using Unbiased Molecular Dynamics simulationPLOS ONE

Dear Dr. Aryapour,

Thank you for submitting your manuscript to PLOS ONE. After careful consideration, we feel that it has merit but does not fully meet PLOS ONE’s publication criteria as it currently stands. Therefore, we invite you to submit a revised version of the manuscript that addresses the points raised during the review process.

We look forward to receiving your revised manuscript.

Kind regards,

Sandipan Chakraborty

Academic Editor

PLOS ONE

Journal Requirements:

Reviewers' comments:

Reviewer's Responses to Questions

**Comments to the Author**

1. Is the manuscript technically sound, and do the data support the conclusions?

Reviewer #1: Yes

Reviewer #2: Partly

Reviewer #3: Partly

2. Has the statistical analysis been performed appropriately and rigorously? 

Reviewer #1: Yes

Reviewer #2: N/A

Reviewer #3: No

3. Have the authors made all data underlying the findings in their manuscript fully available?

Reviewer #1: Yes

Reviewer #2: Yes

Reviewer #3: Yes

4. Is the manuscript presented in an intelligible fashion and written in standard English?

Reviewer #1: Yes

Reviewer #2: Yes

Reviewer #3: Yes

5. Review Comments to the Author

Reviewer #1: Comments;

1. Abstract is very poorly written. Rewrite the abstract according to findings.

2. Clearly define the aim and objectives of the study in the last paragraph of the introduction section.

4. In the Introduction section the author should refer to the research paper and comment on recent in-silico techniques. It will be good information for the readers. I would like to recommend several papers, among many others, providing further explanation on this topic:PMID: 27194485 PMID: 33749525 PMID: 32448055 PMID: 31980008 PMID: 33065246 PMID: 32447145 PMID:  34717229

4. Authors have advised redrawing all the interaction energy graphs in reverse order.

5. The interaction energy values appear to be much larger than those of typical small molecule inhibitors. I think the interaction energy is insufficient to determine the unbinding mode of the ligand accurately. Is there a possibility of overestimation such as incorrectness of reweighing or insufficient sampling of the unbound state?

6. Authors have provided insufficient data. Authors have advised to perform binding studies first before exploring the unbinding.

7. Provide a figure showing the unbinding along with the simulation time.

8. Provide pull parameters in the methods section. Also, briefly explain the general theory of Supervised Molecular Dynamics. How the molecules are pulled out of the pocket, in what direction were the ligands pulled, etc.

9. Define abbreviations in the Abstract section.

10. Why OPLS all-atom force field was used?

11. Authors have not provided the data of external pulling force and contacts to show the unbinding pathway of the selected molecules.

12. Results and discussion lacks data and needs to be elaborated in comparison with other computational data based on similar studies.

13. Authors have advised checking the quality of the minimized structure of the 1HTLV-1 and the authors should carry out additional docking studies with experimentally known inhibitors and compare the computational inhibition values with the experimental values. Without any experimental support or validation studies, the in silico binding free energy calculations may lead us totally wrong results, and the whole work may be nonsense.

14. Overall, the study is incomplete and requires more robust analyses to validate the findings experimentally or computationally. In addition, the manuscript is scientifically unsound and not suitable to publish in this journal without properly validated studies.

Few minor comments;

“We had two mp forms of the potent compound (AspH32 and AspH32′) like the weak compound. In this regard, in the duration times of 4.4us” Make corrections to the unit of time.

What is mp state? The authors have advised to provide abbreviations at their first use.

“In the weak inhibitor unbinding pathways, Trp98 and Trp98′ with pi-pi interactions, due to their close position to one of the exit areas were not good supporters for Asp32 or Asp32′, like His66′, His66′,” Author should correct the notation of His residue.

Authors have calculated the total interaction energy of compound 9 and 10 with htlv but did not mention anything about it in the text. Merely displaying in figures does not make sense.

“We had two mp forms of the potent compound (AspH32 and AspH32′) like the weak compound.” Should mention properly which is potent and each compound.

Authors already mentioned that one compound is better than other and they are just investigating the mechanism of unbinding. In MS authors tells about two mp form of compounds (potent and weak). Is the mp forms be taken into account for getting Ki values for these compounds?

In my view, the results obtained in this study are worthy for publication. The manuscript needs major essential revision before publication. I would like to overview the revised version of the manuscript before it accept for publication.

Reviewer #2: 1. What purpose do authors have selected for two different structures of HTLV-1 protease for this study?

2. Authors have used co-crystalized ligands in this study. What is the novelty of the proposed study?

3. From the literature it was reported that the amino acid Met37 played a key role in the binding of inhibitors in the active site of HTLV-1 protease. Did authors have noticed interaction with this amino acid in this study?

4. Authors have mentioned the uniform mechanism to unbind with some important differences: what are key features or differences noticed during the simulation period?

5. Authors have mentioned that compound shows 526 times more potent in complexes with HTLV-1 than compound 9, is the substitution of amino group of this compound enhances the activity of compound 10?

6. Does the amino group of the compound 10 show any interaction with the important amino acid residues of the HTLV-1 protease?

7. Authors have mentioned that the positive charge of the pyrrolidine ring plays a crucial role in preserving ligands. Both compound 9 and 10 have pyrrolidine rings in their structure, then how compound 9 shows weaker activity than compound 10? Does any specific mechanism play a role in activity of compound 10?

8. In the main text it was mentioned that the figure 4 c-e was the first state of rep1, 2, and 3, but in figure it was mentioned as ligand RMSD. Make it correct.

9. In figure 5b and c, both lines were depicted as Rep2 and Rep3-aniline-benzene. Then how does it show the difference in the distance? What are the main differences noticed in these structures?

10. Is the author using compound 9 as the weaker inhibitor in this study?

Reviewer #3: The manuscript is an in-depth study but needs extensive revisions for publications. Statistical analyses or any convergence tests are missing. It is well written, but figures need to make much better to look like as publishable format. Details review have been attached in the 'reviewer's feedback.docx' file.

6. PLOS authors have the option to publish the peer review history of their article (what does this mean?). If published, this will include your full peer review and any attached files.

Reviewer #1: No

Reviewer #2: No

Reviewer #3: No

---

## [Author Response · Author response to Decision Letter 0]

28 Apr 2022

Dear Prof. Sandipan Chakraborty;

I am pleased to mail you the revised article titled "Comparative study of the unbinding process of some HTLV-1 protease inhibitors using Unbiased Molecular Dynamics simulations". All comments have been answered upon reviewers' recommendations, and the corrections were made in manuscript # PONE-D-22-00753. It is our pleasure to hear your feedback and any other suggestions relevant to the paper.

Sincerely,

Hassan Aryapour

Reviewer #1:

Comment #1:

The abstract is very poorly written. Rewrite the abstract according to findings.

Answer #1:

The abstract section was improved and rewritten.

Comment #2:

Clearly define the aim and objectives of the study in the last paragraph of the introduction section.

Answer #2:

The requested information was added in the last paragraph of the introduction section (lines 100-104)

Comment #3:

In the Introduction section the author should refer to the research paper and comment on recent in-silico techniques. It will be good information for the readers. I would like to recommend several papers, among many others, providing further explanation on this topic: PMID: 27194485 PMID: 33749525 PMID: 32448055 PMID: 31980008 PMID: 33065246 PMID: 32447145 PMID: 34717229

Answer #3:

Some of your mentioned references were added as an example of some in silico methods (lines 72-79)

Comment #4: 

Authors have advised redrawing all the interaction energy graphs in reverse order.

Answer #4:

In this study, we examined unbinding pathways of mentioned inhibitors, so at first, the inhibitors were in a binding position with the highest amount of interaction energies. During the simulation, the weakening of protein-ligand interactions occurred. The inhibitors started to leave the binding site and eventually full unbind occured when the total amount of protein-ligand interaction energies reached zero. So all the interactions energies plots' trend is correlated with the unbinding process from the bound state to the unbound state. This kind of analysis also used in the other studies:

https://doi.org/10.1371/journal.pone.0263251,https://doi.org/10.1371/journal.pone.0257916
https://doi.org/10.1371/journal.pone.0251910

Comment #5:

The interaction energy values appear to be much larger than those of typical small molecule inhibitors. I think the interaction energy is insufficient to determine the unbinding mode of the ligand accurately. Is there a possibility of overestimation such as incorrectness of reweighing or insufficient sampling of the unbound state?

Answer #5:

Total interaction energies plots are obtained from the sum of the coulomb and Lennard-jones contribution values, so the total energies are high. To better understand the complementary analysis, we calculate the average of the most important interaction energies (the amount of these plots is not much as the Total interaction energies plots).

Investigating the energy plots was not the only analysis for determining unbinding processes. We also used the free energy landscape plots obtained using "gmx sham", based on the ligand and protein RMSD values. Eventually, all preliminary information was investigated with trajectory files to advanced details obtain. 

Comment #6:

The authors have provided insufficient data. Authors have advised to perform binding studies first before exploring the unbinding.

Answer #6:

Along with investigating the unbinding pathways of these two inhibitors we also investigated the binding pathways of these inhibitors that we are still writing the article. The information gathered in the binding process has covered the results of the unbinding process. Also there some valuable research regarding only unbinding pathways investigation such as: DOI: 10.1038/srep11539 , DOI: 10.1126/sciadv.1700014, DOI: 10.1073/pnas.1424461112.

Comment #7:

Provide a figure showing the unbinding along with the simulation time.

Answer #7:

In the Table of Contents graphic section, we provided a schematic picture of one of the compound 9 and 10 unbinding pathways as an example. RMSD values of the ligand from binding to complete unbinding positions are also available ( Fig4 a,b). RMSD plots show how ligand location changes with time as compared to their crystallographic binding poses.

Comment #8:

Provide pull parameters in the methods section. Also, briefly explain the general theory of Supervised Molecular Dynamics. How the molecules are pulled out of the pocket, in what direction were the ligands pulled, etc.

Answer #8:

We use a simulation method (SuMD) entirely different from “umbrella sampling” to pull ligands from their binding sites. A brief explanation of the SuMD method is available on lines 91-97. The directions of inhibitor unbinding are available in the Result and Discussion section, for example: " In rep2 and 3, the inhibitor unbounded between semi-open flaps forms (Figure 14e, 14f) (Figure 15a, 15b, 15c)". 

Comment #9:

Define abbreviations in the Abstract section.

Answer #9:

The requested correction was done in the Abstract section (line 25).

Comment #10:

Why OPLS all-atom force field was used?

Answer #10:

Because we used the OPLS force field for binding simulations, the unbinding simulations were also done by it to reproduce the data. Also, in some other similar research involving: https://doi.org/10.1093/bioinformatics/btaa565, DOI: 10.1126/sciadv.1700014, this force field is used. In OPLS, surfaces (parameters) are fit to experimental data sets MM PES, so this FF is perfect for describing the properties of small molecules in their condensed states. OPLS-AA is entirely open-source, so it can be applied in many MD programs; particularly, the most popular MD program has already integrated the OPLS-AA into it.

Comment #11:

Authors have not provided the data of external pulling force and contacts to show the unbinding pathway of the selected molecules.

Answer #11:

The process of unbinding from binding pose to the full unbind is dependent on a set of factors, including aspects of protein-ligand interactions, water mediation, protein, and ligand fluctuation. In the Result and Discussion section, we explained the atomistic details of all of these factors with different analyses. As we answered in “Comment#8” we did not use pull parameters.

Comment #12:

Results and discussion lack data and need to be elaborated in comparison with other computational data based on similar studies.

Answer #12:

In the other computational research, the study's goal differed. Some of our data were new, so they could not be compared, except our other team's research that works on unbinding pathways of Indinavir in complex with HTLV-1 protease. Previously, we discussed our correlation of results in the conclusion section "These residues' interactions are unfavorable to Indinivar's stability in complex with HTLV-1 protease and result in it being a weak inhibitor".

Comment #13:

Authors have advised checking the quality of the minimized structure of the 1HTLV-1 and the authors should carry out additional docking studies with experimentally known inhibitors and compare the computational inhibition values with the experimental values. Without any experimental support or validation studies, the in silico binding free energy calculations may lead us totally wrong results, and the whole work may be nonsense.

Answer #13:

In the other project, we investigated the binding pathways of the two inhibitors, which will be available soon. By combining the results of this article and binding study results, we can have a more positive attitude towards drug design. 

Only crystallographic binding pose information is analyzed in the experimental research, but we study the interactions during the pathways. As you run more simulations, you will discover more pathways. However, we added the results compared with experimental research in the conclusion section (lines 433-434 and 460-467) to better understand.

Few minor comments;

We had two mp forms of the potent compound (AspH32 and AspH32′) like the weak compound. In this regard, in the duration times of 4.4us" Make corrections to the unit of time.

What is mp state? The authors have advised to provide abbreviations at their first use.

"In the weak inhibitor unbinding pathways, Trp98 and Trp98′ with pi-pi interactions, due to their close position to one of the exit areas were not good supporters for Asp32 or Asp32′, like His66′, His66′," Author should correct the notation of His residue.

Authors have calculated the total interaction energy of compound 9 and 10 with htlv but did not mention anything about it in the text. Merely displaying in figures does not make sense.

"We had two mp forms of the potent compound (AspH32 and AspH32′) like the weak compound." Should mention properly which is potent and each compound.

Authors already mentioned that one compound is better than other and they are just investigating the mechanism of unbinding. In MS authors tells about two mp form of compounds (potent and weak). Is the mp forms be taken into account for getting Ki values for these compounds?

Answer:

A unit correction was done. The abbreviation of mp was added in the Abstract section. The correction was done in "In the weak inhibitor unbinding pathways, Trp98 and Trp98′ with pi-pi interactions, due to their close position to one of the exit areas were not good supporters for Asp32 or Asp32′, like His66′".

Total interaction energies plots are obtained from the sum of the coulomb and Lennard-jones contribution values. We have maximum interaction energies in the bound state of the protein-ligand complex. As time passes and protein-ligand interactions weaken, the values of the plot increase and eventually reach zero until full unbinding. So all the interactions energies plots' trend is correlated with the unbinding process from the bound state to the unbound state. So we used these plots to prove full unbind occurrence by energy values.

Aspartic proteases tend to cleave substrate by monoprotonation of aspartates in the catalytic dyad. Since HIV protease is an aspartic protease, and unfortunately, there was no typical study about HTLV-1 protease, we decided to take a pattern from HIV research. So in some research, they do not apply protonation or do not mention it, or in some research, they apply protonation state in chain A. Due to we wanted to have more complete outcome, we decided to use protonation in both chains separately. 

We mentioned in the conclusion section, "Herein, we cannot conclude certainly which state of protonation actually occurs". In the presence of an inhibitor, the kind of protonation that will occur can only be determined when exposed to the active site. Because of this, we decided to discuss protonation more cautiously and, instead of judging, report the results.

Reviewer #2:

Comment #1:

What purpose do authors have selected for two different structures of HTLV-1 protease for this study?

Answer #1:

As we mentioned in the Result and Discussion section, "Since the only structural difference between compounds 9 and 10 is in the amino and nitro groups on the benzene ring (Figure 1a, 1b), compound 10 (Ki = 15 nM) is approximately 526 times more potent in complex with HTLV-1 protease. Therefore, a proper understanding of the unbinding pathways of these compounds is vital to unveiling secrets that a minor structural difference can have a dramatic effect on inhibitory effects." We could monitor critical factors from a comparative perspective, which made the results more reliable for designing new inhibitors. The related section improved in the manuscript.

Comment #2:

The authors have used co-crystallized ligands in this study. What is the novelty of the proposed study?

Answer #2:

There is no FDA-approved drug for this target, so we decided to work on designed inhibitors. In our view, it is critical to determine the negatives and the positives of the current potent inhibitor. We also compared the result with a weak inhibitor to achieve a complete insight for designing next-generation inhibitors.

Comment #3:

From the literature, it was reported that the amino acid Met37 played a key role in the binding of inhibitors in the active site of HTLV-1 protease. Did authors have noticed interaction with this amino acid in this study?

Answer #3:

In our replicas, Met37 effects were not significant as critical residues. Important interaction energies values are available in energy plots.

Comment #4:

Authors have mentioned the uniform mechanism to unbind with some important differences: what are key features or differences noticed during the simulation period?

Answer #4:

The Phrase "we saw a uniform mechanism to unbind with some important differences that caused a significant difference in one of simulation time" was used for compound 10 unbinding pathways in complex with HTLV-1 protease when Asp32 of chain A was protonated. We mentioned "In this state, Lys95′ by forming a hydrogen bond with the atom of O10, along with His66′ cation-pi interaction with the benzene ring, was a third essential residue" (lines 231-232) and "According to significant differences in replicas simulation times, the effects of the Lys95′ hydrogen bond (Figure 4i) appear more pronounced" (lines 235-236) as a key factor that caused significant different in simulation time in the Result and Discussion section. 

Comment #5:

Authors have mentioned that the compound shows 526 times more potent in complexes with HTLV-1 than compound 9, is the substitution of the amino group of this compound enhances the activity of compound 10?

Answer #5:

Yes, it is. The inhibitor 10 could be able to interact with His66′ and Lys95′, which are essential.

Comment #6:

Does the amino group of the compound 10 show any interaction with the important amino acid residues of the HTLV-1 protease?

Answer #6:

Yes, it does. Asp36′ by forming H-bond when Asp32 of chain A was protonated, and Asp36 and Asp36′ by forming H-bonds when Asp32 of chain B was protonated with the amino group, which is available in the text line 226 and line 270-274.

Comment #7:

Authors have mentioned that the positive charge of the pyrrolidine ring plays a crucial role in preserving ligands. Both compound 9 and 10 have pyrrolidine rings in their structure, then how compound 9 shows weaker activity than compound 10? Does any specific mechanism play a role in activity of compound 10?

Answer #7:

We mentioned in the Conclusion section, "The pyrrolidine fragment was held well by Asp32 or Asp32′ from the native binding pose of the two compounds. Thus the interactions of other fragments with other residues in different protein regions caused significant differences." Moreover, "His66′ with its cation–pi interaction with an aniline ring of inhibitor was a perfect supporter to Asp32′. This residue's effect was absent in the other form of protonation state and caused a significant difference in simulation time. In the weak inhibitor unbinding pathways, Trp98 and Trp98′ with pi-pi interactions, due to their close position to one of the exit areas, were not good supporters for Asp32 or Asp32′, like His66′. His66′, due to far position from the bottom and center of the binding pocket, could fixed aniline fragment and decreased ligand fluctuations. The two mentioned tryptophan were closer to the important aspartic acids, and there was enough space for ligand fluctuations. For this reason, Asp36 in the active site that was close to the exit area could be a competitor with Asp32′ and was not a good interaction for keeping the ligand in the binding pocket. Similarly, attenuating effect of Trp98/Trp98′ residues in unbinding pathways of the weak inhibitor correlates with another research result" https://doi.org/10.1371/journal.pone.0257916

Comment #8:

In the main text it was mentioned that the figure 4 c-e was the first state of rep1, 2, and 3, but in figure it was mentioned as ligand RMSD. Make it correct.

Answer #8:

In Figure 4 , ligand RMSD values are for 4a and 4b.

Comment #9:

In figure 5b and c, both lines were depicted as Rep2 and Rep3-aniline-benzene. Then how does it show the difference in the distance? What are the main differences noticed in these structures?

Answer #9:

These plots aim to show and prove pi-pi intramolecular interactions in the inhibitors. So we considered the two aniline-benzene rings in compound 10 and the two nitrobenzene-benzene rings in compound 9, which were closer together to make intramolecular interactions. These plots should not compare with each other.

Comment #10:

Is the author using compound 9 as the weaker inhibitor in this study?

Answer #10:

Yes 

Reviewer #3:

Comment #1:

After lots of simulations and analyses, it is still unclear that which states of protonation plays the deciding role there. It would be better if the authors are critical on this point because it is one of the most important viewpoints of this work.

Answer #1:

According to all forms of protonation, there are 16 forms. A QM/MM study will help us determine which protonation forms are closer to the natural form. Also, Aspartic proteases tend to cleave substrate by monoprotonation of aspartates in the catalytic dyad. Since HIV protease is an aspartic protease, and unfortunately, there was no typical study about HTLV-1 protease, we decided to take a pattern from HIV research. So in some research, they do not apply protonation or do not mention it, or in some research, they apply protonation state in chain A. Due to we wanted to have a complete outcome, we decided to use protonation in both chains separately. 

Comment #2:

Authors in many places discussed about pi-cation and pi-anion interaction (e.g., His66 interacts with aniline in cation-pi interaction, Asp36 forming pi-anion interaction with nitrobenzene etc.). It is important to note that the energetics of such interactions are very important. (For example, pi-anion interactions have energetics in the range around 20 to 50 kj/mol whereas pi-cation interactions have energetics in the range of -10 to -20 kj/mol). The authors should check the energetics of such interactions here or at least cite relevant papers where people have reported the energetics of these particular system.

Answer #2:

We first reported the interaction energy values, so we cannot compare them or cite other works. We chose significant interactions based on the average amount of most important interaction energies of the protein-ligand complex. Our study values are negative and favorable since the nitro group is a strong electron acceptor and positively attaches to the ring. It can easily be made by pi-anion interaction with aspartic acid, which carries a negative charge.

Comment #3:

Authors should rigorously work on the figure qualities. Those are not good at publication format presently. For example, Figure 13, c, d, e. Note that, the labellings are very bold and not as per the mark of publication. Also, in 13a, the x-axis labellings are very closely spaced (10,20,30,40….80) makes it looks worse. They should scrutinize the figure qualities and font labelling throughout in-depth.

Answer #3:

The requested changes were applied to the figures.

Comment #4:

In Figure 10 caption (where they emphasize the details of compound 9 unbinding pathway), just before the last line, they mentioned "k, l, 3", What is "3" here? I did not find anything such and believe "3" should be replace by "m". Such things they should check thoroughly.

Answer #4:

The corrections were done.

Comment #5:

It is better to follow a uniform free energy bar throughout which is currently missing. For example, Figure 7, c, d, e. Their energy bars are not uniform (For c, it is 0 to 16.90 and it is different for d and e.). The authors should follow uniform energy bar for all, otherwise the comparison is not at all logical.

Answer #5:

The FEL plots were prepared by python, and the python codes made the energy bars automatically from different values obtained using "gmx sham" files in different replicas. Also, we tried uniforming plots scale, but some of the plots became too compact and became difficult to separate steps.

Comment #6:

The authors mention that "For our purpose, we had various short and long-time-scale simulations". I understand they chose based on the unbinding time scales, still did the authors performed any convergence tests or something like that, to choose time scales in this study?

Answer #6:

We did not do that. Because all full unbound occurred automatically without applying any biasing force. Various simulation times showed various unbinding processes. We corrected the text to prevent misinterpretation ( line 208). 

Comment #7:

Figure 5, for the three replica's they chose three different time scales, that should be addressed properly. What happened if for Figure 5b and 5c, they did 4000 ns simulations like Figure 5a? Then, how the distance plot would look like? Since, they are doing unbiased simulations, these questions need to be answered. Also, it is visually more sound to have similar time-scales for different replica's.

Answer #7:

The time it takes for an inhibitor to leave its binding pocket depends on the pathway it is sampled. Unbinding pathways may contain multiple energy barriers, so the residence time of the inhibitor at the binding site increases, and it leaves the protein later. The inhibitor or replica whose unbinding pathway has fewer energy barriers exits the protein faster. All replicas are simulated under identical conditions using SuMD.

Comment #8:

Figure 3g, take care of the x axis label. It has been cut-off (120 ns is cut-off there). Same goes for Figure 8c. Also, The TOC (as well as TOC fonts) is not looking good as per the journal standard.

Answer #8:

The correction was done, and also revised version of the article was changed to meet PLOS ONE's style requirements.

Comment #9:

The article is somewhat looks to be less referenced. For an example, they should cite some important references regarding the interactions such as the review articles sastry et al doi: 10.1021/cr300222d and doi: 10.1021/jp900013e, and some others, for example https://doi.org/10.1021/ct200569x,https://doi.org/10.1021/acs.jpcb.7b01736and
https://doi.org/10.1021/acs.jpcb.9b06343 etc.

Answer #9:

Some references were added in lines 224 and 340.

---

## [Decision Letter · Decision Letter 1]

23 May 2022

PONE-D-22-00753R1Comparative study of the unbinding process of some HTLV-1 protease inhibitors using Unbiased Molecular Dynamics simulationsPLOS ONE

Dear Dr. Aryapour,

Thank you for submitting your manuscript to PLOS ONE. After careful consideration, we feel that it has merit but does not fully meet PLOS ONE’s publication criteria as it currently stands. Therefore, we invite you to submit a revised version of the manuscript that addresses the points raised during the review process.

We look forward to receiving your revised manuscript.

Kind regards,

Sandipan Chakraborty

Academic Editor

PLOS ONE

Journal Requirements:

Reviewers' comments:

Reviewer's Responses to Questions

**Comments to the Author**

1. If the authors have adequately addressed your comments raised in a previous round of review and you feel that this manuscript is now acceptable for publication, you may indicate that here to bypass the “Comments to the Author” section, enter your conflict of interest statement in the “Confidential to Editor” section, and submit your "Accept" recommendation.

Reviewer #1: All comments have been addressed

Reviewer #2: All comments have been addressed

Reviewer #3: (No Response)

2. Is the manuscript technically sound, and do the data support the conclusions?

Reviewer #1: Yes

Reviewer #2: Yes

Reviewer #3: Partly

3. Has the statistical analysis been performed appropriately and rigorously? 

Reviewer #1: Yes

Reviewer #2: N/A

Reviewer #3: N/A

4. Have the authors made all data underlying the findings in their manuscript fully available?

Reviewer #1: Yes

Reviewer #2: Yes

Reviewer #3: Yes

5. Is the manuscript presented in an intelligible fashion and written in standard English?

Reviewer #1: Yes

Reviewer #2: Yes

Reviewer #3: Yes

6. Review Comments to the Author

Reviewer #1: The authors have responded to all concerns meticulously and improved the manuscript accordingly. The revised draft is improved significantly. I do not have further comments.

Reviewer #2: The authors have responded all the queries by reviewers, and in present form the manuscript can be accepted for the publications.

Reviewer #3: (No Response)

7. PLOS authors have the option to publish the peer review history of their article (what does this mean?). If published, this will include your full peer review and any attached files.

Reviewer #1: No

Reviewer #2: No

Reviewer #3: No

---

## [Author Response · Author response to Decision Letter 1]

9 Jun 2022

Dear Prof. Sandipan Chakraborty;

I am pleased to mail you the second revision of the article titled "Comparative study of the unbinding process of some HTLV-1 protease inhibitors using Unbiased Molecular Dynamics simulations". All comments have been answered upon reviewers' recommendations, and the corrections were made in manuscript #PONE-D-22-00753R1. It is our pleasure to hear your feedback and any other suggestions relevant to the paper.

Sincerely,

Hassan Aryapour

Comment #1:

The author mentioned to my first query that “So in some research, they do not apply protonation or do not mention it, or in some research, they apply protonation state in chain A” …. It would be good if they discuss those studies briefly in the revised manuscript to get the reader a subtle grasp about the history of the study which is still somewhat missing.

Answer #1:

The requested description was added along with the reference (lines 113-117)

Comment #2:

According to my comment 5 regarding uniform free energy bar…. I think it’s needed (Even if not in the manuscript, then in the SI) because that not only establish the fidelity of the work but also required for proper comparison between the states. This is because, we know the free energy changes dramatically based on scaling and non-uniformity of the values selected.

Answer #2:

Yes, it is definitely easier to compare between states if the maximum values on the x and y scale are the same in all graphs. So once again, according to your request, we considered the maximum values of the x and y axes to be the same. Unfortunately, since in some replicas, the amount of maxima is significantly different from each other, a lot of space on the plots becomes empty and white, so the maximum amount of axes is automatically adjusted based on the maximum amount of data after running the script. However, the changes you requested were added to the supplementary section.

Comment #3:

I am somewhat surprised regarding the revision of the author of my comment 9. They did not incorporate none of the suggested studies and just incorporated two different references. Those suggested literature evidence are important and need to be incorporated.

Answer #3:

All the requested references were added (refs : 18, 38,39).

---

## [Editor Report · Decision Letter 2]

16 Jun 2022

PONE-D-22-00753R2Comparative study of the unbinding process of some HTLV-1 protease inhibitors using Unbiased Molecular Dynamics simulationsPLOS ONE

Dear Dr. Aryapour,

Thank you for submitting your manuscript to PLOS ONE. After careful consideration, we feel that it has merit but does not fully meet PLOS ONE’s publication criteria as it currently stands. Therefore, we invite you to submit a revised version of the manuscript that addresses the points raised during the review process.

We look forward to receiving your revised manuscript.

Kind regards,

Sandipan Chakraborty

Academic Editor

PLOS ONE

**Additional Editor Comments: **

Authors successfully address the reviewer concerns. However, the following issue needs to be clarified before any final decision on the manuscript.

The author of the manuscript entitled "Comparative study of the unbinding process of some HTLV-1 protease inhibitors using Unbiased Molecular Dynamics simulations" performed ligand unbinding from the protein HTLV-1. They consider one high affinity ligand and a low affinity ligand. However, the author published a previous paper on "Comparative analysis of the unbinding pathways of antiviral drug Indinavir from HIV and HTLV1 proteases by supervised molecular dynamics simulation" where they consider unbinding of another ligand Indinavir. The method is very similar and presentation is also very similar. A small discussion is there in the manuscript. But the author needs to justify their objective to work on different ligands and clearly define the novelty of the work in light of previous publication.
---

## [Author Response · Author response to Decision Letter 2]

27 Jun 2022

Dear Prof. Sandipan Chakraborty;

I am pleased to mail you the second revision of the article titled "Comparative study of the unbinding process of some HTLV-1 protease inhibitors using Unbiased Molecular Dynamics simulations". All comments have been answered upon reviewers' recommendations, and the corrections were made in manuscript #PONE-D-22-00753R1. It is our pleasure to hear your feedback and any other suggestions relevant to the paper.

Sincerely,

Hassan Aryapour

Comment:

Authors successfully address the reviewer concerns. However, the following issue needs to be clarified before any final decision on the manuscript.

The author of the manuscript entitled "Comparative study of the unbinding process of some HTLV-1 protease inhibitors using Unbiased Molecular Dynamics simulations" performed ligand unbinding from the protein HTLV-1. They consider one high affinity ligand and a low affinity ligand. However, the author published a previous paper on "Comparative analysis of the unbinding pathways of antiviral drug Indinavir from HIV and HTLV1 proteases by supervised molecular dynamics simulation" where they consider unbinding of another ligand Indinavir. The method is very similar and presentation is also very similar. A small discussion is there in the manuscript. But the author needs to justify their objective to work on different ligands and clearly define the novelty of the work in light of previous publication.

Answer:

We said in lines 59-69 that many research groups had designed inhibitors, but the FDA has approved none. For this reason, if we want to design successful inhibitors, we must examine existing inhibitors' negative and positive points. These will ultimately lead to valuable information for drug design.

In a previous project, our research team discovered the reasons for Indinavir's ineffectiveness against HTLV-1 protease. As we continue, we examined the unbinding pathways of two inhibitors (the most potent and one of the weakest) in complex with HTLV-1 protease. Although the study method was the same, the aim of the research was different since, in the previous project, we had one drug and two different target proteins, while in the present study, we have two different inhibitors for only one target protein.

More information was added in the Introduction section (lines 98-100), and the common results of these two projects were mentioned before in lines 456-457.

---

## [Editor Report · Decision Letter 3]

28 Jun 2022

Comparative study of the unbinding process of some HTLV-1 protease inhibitors using Unbiased Molecular Dynamics simulations

PONE-D-22-00753R3

Dear Dr. Aryapour,

We’re pleased to inform you that your manuscript has been judged scientifically suitable for publication and will be formally accepted for publication once it meets all outstanding technical requirements.

Kind regards,

Sandipan Chakraborty

Academic Editor

PLOS ONE

---

## [Editor Report · Acceptance letter]

4 Jul 2022

PONE-D-22-00753R3 

Comparative study of the unbinding process of some HTLV-1 protease inhibitors using Unbiased Molecular Dynamics simulations 

Dear Dr. Aryapour:

I'm pleased to inform you that your manuscript has been deemed suitable for publication in PLOS ONE. Congratulations! Your manuscript is now with our production department. 

Kind regards, 

on behalf of

Dr. Sandipan Chakraborty 

Academic Editor

PLOS ONE